# Accuracy of Estimated Fetal Weight Assessment in Fetuses with Congenital Diaphragmatic Hernia—Is the Hadlock Formula a Reliable Tool?

**DOI:** 10.3390/jcm13123392

**Published:** 2024-06-10

**Authors:** Daria Kuchnowska, Albert Stachura, Przemyslaw Kosinski, Maciej Gawlak, Piotr Wegrzyn

**Affiliations:** 1Department of Obstetrics, Perinatology and Gynecology, Medical University of Warsaw, 02-091 Warsaw, Poland; daria.kuchnowska@gmail.com (D.K.); maciej.gawlak1@wum.edu.pl (M.G.); piotr.wegrzyn@wum.edu.pl (P.W.); 2Center for Preclinical Research, Department of Methodology, Doctoral School, Medical University of Warsaw, 02-091 Warsaw, Poland; albert.stachura@wum.edu.pl

**Keywords:** congenital, diaphragmatic, hernia, fetal weight, prenatal diagnosis

## Abstract

**Objectives**: Congenital diaphragmatic hernia (CDH) is defined as organ protrusion from the abdominal to the thoracic cavity. The Hadlock formula is the most commonly used tool for calculating estimated fetal weight (EFW). The anatomical nature of CDH usually leads to underestimation of the abdominal circumference, resulting in underestimation of fetal weight. Accurate weight estimation is essential before birth for counselling, preparation before surgery and ECMO. The research is made to compare the accuracy of Hadlock’s formula and Faschingbauer’s formula for fetal weight estimation in CDH fetuses population. **Methods**: In our study, we investigated differences between EFW and actual birthweight in 42 fetuses with CDH as compared to 80 healthy matched controls. EFW was calculated using the Hadlock formula and a recently introduced formula described by Faschingbauer et al., which was tailored for fetuses with CDH. Additionally, both of the formulas were adjusted for the interval between the ultrasound and delivery for both of the groups. **Results**: The majority of hernias were left-sided (92.8% vs. 7.2%). EFW adjusted for the interval between the ultrasound and delivery had the highest correlation with the actual birthweight in both, study group and controls. We compared the results for both tools and found the Hadlock formula to predict birthweight in CDH children with a 7.8 ± 5.5% error as compared to 7.9 ± 6.5% error for the Faschingbauer’s formula. **Conclusions**: The Hadlock formula adjusted for the interval between the ultrasound and delivery is a more precise method of calculating EFW in fetuses with CDH. Routine biometry scan using Hadlock’s formula remains reliable for predicting birthweight.

## 1. Introduction

Congenital diaphragmatic hernia (CDH), whose incidence has been estimated at approximately 1 in 3000 pregnancies [1,2], is a malformation of the diaphragm which leads to organ herniation from the abdomen to the thorax. Posterolateral defects on the left side of the diaphragm comprise most cases of diaphragmatic hernia. The defect is usually unilateral—left-sided in 80% and right-sided in 13% of the cases, with bilateral hernia reported in only 2% of the cases. Congenital diaphragmatic hernia (CDH) is a non-homogenous anatomical defect, ranging from extremely large defects in the diaphragm, with 90–100% mortality rates, to slight defects of little clinical significance, with 90–100% survival rates among the affected infants [2].The prognosis depends on the size of the defect and the degree of pulmonary hypoplasia, which in turn depends on the affected side (left or right) and which organs herniated into the thoracic cavity. In congenital diaphragmatic hernia lung capacity is decreased, causing complications such as pulmonary hypoplasia and pulmonary hypertension, which is associated with elevated risk for neonatal death. Prenatal ultrasonography provides important clinical information, which may be useful in neonatal management planning and, in selected cases, prenatal treatment would not be possible without accurate diagnosis [3].

Routine biometric assessment of the developing fetus, including cases with CDH, allows to estimate fetal weight and plan both, time and mode of delivery. It is also used to calculate perinatal risk for fetuses with and without CDH [4]. Furthermore, estimated fetal weight (EFW) plays a key role in planning extracorporeal membrane oxygenation (ECMO), as infants with EFW <2000 g are not eligible for this procedure at most centers [5]. Estimated fetal weight is commonly calculated using the Hadlock’s three-parameter (HC, AC and FL) formula (EFW = 10^(1.326 + 0.0107 × HC + 0.0438 × AC + 0.158 × FL − 0.00326 × AC × FL)), which is associated with the least error in estimating EFW [6]. The accepted range of error for the Hadlock formula is up to 20% in either direction for EFW accuracy [7]. However, in fetuses with CDH, the error margin may be higher because of the decreased abdominal circumference due to bowel, stomach and/or liver displacement into the fetal thorax. The prognostic influence of prenatally diagnosed liver herniation in the hemithorax in fetuses with CDH has been widely discussed and recognized as a significant marker of neonatal survival [8]. The most severe cases of CDH are offered fetoscopic endotracheal occlusion (FETO) in some centers [9]. Recent randomized studies have demonstrated this procedure to be effective in increasing survival of the most severe cases [10,11].

In this study, we assumed that the estimated fetal weight in fetuses with CDH would be lower due to reduced abdominal circumference as compared to healthy babies. In 2015, an alternative EFW formula for calculating fetal weight in cases with CDH was described by Faschingbauer et al. [12]. Some recent studies have suggested that CDH-specific formula for calculating EFW is more accurate than the Hadlock formula in this population [13]. In our study, we compared the Hadlock’s formula and a Faschingbauer’s formula, designed especially for fetuses with CDH, to determine whether Hadlock’s formula is a reliable tool in a group of patients with isolated CDH [12,14]. Furthermore, we compared the results and error margins to a control group where the error is not caused by the migration of organs to the chest cavity.

## 2. Materials and Methods

This retrospective case-control study was conducted at the Department of Obstetrics, Perinatology and Gynecology, Medical University of Warsaw, Poland. The study group included a cohort of fetuses prenatally diagnosed or confirmed with CDH at our center, from October 2017 to April 2021. The control group included healthy pregnant women who gave birth to healthy neonates (no congenital abnormalities, Apgar Score 10 and umbilical cord acid-base parameters within normal ranges). The cases in the study group were matched with controls for the following confounding variables: maternal age, BMI, gravidity, and parity, as well as gestational age at delivery.

We excluded fetuses with chromosomal abnormalities and included only pregnancies with congenital diaphragmatic hernia as an isolated defect, and no major maternal comorbidities. Cases of both, right- and left-sided diaphragmatic hernias were included in the analysis.

The FETO procedure was neither an exclusion nor an inclusion criterion, so long as there was no preterm labor or pPROM, whose occurrence was an exclusion criterion. Gestational age at delivery of <37 weeks was an exclusion criterion. All fetuses were diagnosed with congenital diaphragmatic hernia before delivery. Gestational age was calculated based on the last menstrual period and was confirmed by, or recalculated with, CRL measurements from the first-trimester ultrasound.

The primary outcome of our study was the difference between the estimated fetal weight (EFW) and neonatal birthweight, expressed as an absolute value and percentage. Two methods were used to calculate EFW: the Hadlock formula and a new equation, designed specifically for fetuses with diaphragmatic hernia, and introduced by Faschingbauer [12]. The Faschingbauer’s formula was based on research of 146 cases of congenital diaphragmatic hernia—a retrospective study in which newborns delivered from a singleton pregnancy were measured within 7 days before birth. Afterwards, a graphical analysis of the relationship between the measurements was performed, followed by applying multivariate fractional polynomials to derive the Faschingbauer’s formula. The Faschingbauer’s formula includes measurements of HC, AC and FL, while also accounting for liver herniation. The full formula presents as: EFW = 10^(4.6729 107 371 + 0.2365 011 768 * HC + 0.2228 897 682 * FL2 − 0.0129 895 773 * FL3 − 1.0470 039 072 * (FL * HC)0.5 + 0.0004 314 661 * (AC * HC) − [in case of liver herniation] 0.0062 112 122).The study was cross-checked with other fetal weight estimation formulas and proved the Faschingbauer’s method to be the most accurate tool for predicting fetal weight in newborns with congenital diaphragmatic hernia.

In case of multiple ultrasound examinations, the most recent scan was considered. To account for the time interval between the final growth ultrasound and the actual date of delivery, we used previously described estimates of fetal growth of 185 g per week from 35 to 40 weeks [15]. Therefore, to adjust for the time gap between weight estimation and the actual date of delivery, we used the adjusted EFW formula: adjusted EFW = EFW + 185*time gap [days]/7. The adjusted EFW formula aims to adjust for the EFW-delivery interval and minimize this as a source of error. In this study, five different values were compared: actual birthweight, Hadlock’s formula EFW, Faschingbauer’s formula EFW, adjusted EFW of Hadlock’s formula and adjusted EFW of Faschingbauer’s EFW formula.

For statistical analysis, Shapiro-Wilk test was used to determine the normality of data distribution. Either unpaired *t*-test or Mann-Whitney U test were used to compare data between the two groups. Paired t-test was implemented to calculate differences between EFW and birthweight. Pearson’s correlation coefficient was calculated for actual birthweight and fetal weight using different methods. Friedman ANOVA was used to find differences in percentage weight error between various measurement methods. Later, post-hoc analysis with Wilcoxon test was performed to compare individual measurements. Bonferroni correction for multiple comparisons was used, with statistical significance of *p* < 0.0083 for these calculations. The statistical significance threshold was set at *p* < 0.05, unless otherwise indicated. Data were visualized, using GraphPad Prism 9. Bland-Altman plots were used to establish the 95% limits of agreement for all measurement methods in both studied groups. Calculations and figures were prepared in R version 4.3.0.

Ethical approval was obtained for this study in accordance with local guidelines. Written informed consent from participants was not required in accordance with local guidelines.

## 3. Results

A total of 42 cases of isolated CDH were included in the study. The control group consisted of 80 healthy neonates. At baseline, the two groups differed only by time elapsed between ultrasound scan and delivery (mean difference in days: 0.84 vs. 5.6, *p* < 0.001) (shown in Table 1). The birthweight of CDH neonates was significantly lower as compared to healthy controls. On ultrasound, no significant differences in biparietal diameter, head circumference, percentiles for head circumference and femur length were found. Mean abdominal circumference was lower in CDH newborns as compared to controls (341.1 mm vs. 320.3 mm). Biparietal diameter measurements were higher in CDH cases (92 mm vs. 92.5 mm), although the difference was not statistically significant, and femur length was higher in controls (73.4 mm vs. 70.9 mm). No significant differences in absolute values between the birthweight and EFW or adjusted EFW were observed in the control group.

The results obtained with both, adjusted for time between ultrasound and delivery and unadjusted Faschingbauer’s formula differed substantially from neonatal weight at delivery (shown in Table 2). The majority of hernias were left-sided (92.8% vs. 7.2%). The number of the right-sided congenital diaphragmatic hernias was low, as this type of herniation is less common. The characteristics of ultrasound measurements in fetuses with left-sided vs. right-sided CDH are shown in Table 3.

Out of 42 cases, 9 (21.4%) underwent the FETO procedure. Within the FETO group, only patients with no preterm delivery or premature rupture of membranes were analyzed. Birthweight score was the only statistically significant difference between FETO and no-FETO groups. The actual birthweight was significantly higher in the no-FETO as compared to the FETO group. No differences between the groups were found as far as the rest of measurements were concerned (shown in Table 4).

Adjusted EFW had the highest correlation with the actual birthweight in both, study group and controls. In the study group, predicted and actual weight was significantly different only when using the unadjusted Hadlock formula. We compensated for this discrepancy by adjusting EFW for time elapsed between the ultrasound measurements and delivery. Both, adjusted and unadjusted Faschingbauer’s formula predicted neonatal weight accurately, although a weaker correlation with the actual birthweight than adjusted EFW, calculated with Hadlock formula, was observed (shown in Table 5, Figure 1 and Figure 2).

The mean differences for the study group are: EFW −1.83 ± 290.71 (95% limit of agreement: −572 to 568), adjusted EFW 20.33 ± 286.95 (95% limit of agreement: −542 to 583), Faschingbauer’s formula 141.67 ± 288.37 (95% limit of agreement: −424 to 707), adjusted Faschingbauer’s formula 163.82 ± 283.61 (95% limit of agreement: −392 to 720). The mean differences for the CDH group are: EFW −220.21 ± 325.13 (95% limit of agreement: −857 to 417), adjusted EFW −82.41 ± 296.86 (95% limit of agreement: −664 to 499), Faschingbauer’s formula −59.64 ± 322.79 (95% limit of agreement: −692 to 573), adjusted Faschingbauer’s formula 78.17 ± 297.15 (95% limit of agreement: −504 to 661).

We compared differences in weight between the study group and controls. The results obtained with the unadjusted Hadlock formula were associated with significantly higher error in the study group than in controls (9.7% vs. 6.5%, respectively, *p* = 0.005). Other methods generated comparable results. (shown in Table 5). We also illustrated differences between various measurement methods between the groups (shown in Figure 3). In the study group, the percentage difference in EFW decreased significantly after adjusting for time elapsed between measurements and delivery. In the control group, we noted significant differences in percentage inaccuracy between Hadlock and the Faschingbauer’s formula, irrespective of the adjustments for time elapsed between measurements and delivery (average values are shown in Table 6). As the aim of the study is to compare formulas to accurately estimate fetal weight in fetuses with congenital diaphragmatic hernia, we have also compared formulas in both groups—the CDH group and control group. These data are presented in Table 7 and Table 8.

## 4. Discussion

In this study, we demonstrated that the Hadlock formula is a reliable method for calculating birthweight in infants with CDH, if adjusted for the time interval between the ultrasound measurements and delivery. The use of a novel, tailored approach proposed by Faschingbauer et al., showed no definite advantage over Hadlock within our study sample. The adjusted EFW formula provided the most accurate results for both groups, after adjusting for the time elapsed since ultrasound measurements [16].

We found significant differences in fetal biometry between the groups. CDH infants were on average smaller than their healthy peers, possibly due to poor organ development resulting from herniation. Abdominal circumference (AC) and AC percentiles were significantly lower in CDH infants, which might have been the result of organ migration from the abdominal into the thoracic cavity, but also difference in gestational age at the delivery. According to some authors, abdominal circumference measurement of <5 percentile as a single ultrasound marker in the second trimester as well as birthweight are reliable predictors of poor prognosis in fetuses with CDH [17,18,19].

Faschingbauer et al., conducted a case-control study and found biometrical discrepancies between healthy population and fetuses with CDH [20]. In contrast, Zuckerwise et al., demonstrated the EFW error margin to be within the accepted limits for fetuses with herniation [13]. The results of the abovementioned studies remain conflicting, questioning the reliability of EFW in CDH fetuses.

Rode et al., found no evidence of lower abdominal circumference and EFW in fetuses with CDH as compared to the healthy population, although without cross-checking with a matched control group [21]. Most of the remaining authors suggest that lower AC in CDH fetuses results from visceral organs herniation into the thorax. CDH fetuses with liver herniated into the hemithorax have significantly poorer prognosis than similarly afflicted fetuses without liver herniation [8].

We have also calculated 95% limits of agreement for different formulas in the study and control groups. The results are presented as Bland-Altman plots where 95% limits of agreements are indicated by red dashed lines. In the control group both EFW and adjusted EFW are overestimating and underestimating results within similar limits. Both the Faschingbauer’s formula and adjusted Faschingbauer’s formula have a higher overestimation tendency. Within CDH fetuses the highest risk of EFW underestimation is when a regular Hadlock formula is used and the lowest EFW underestimation was noticed with the use of the new adjusted formula. As shown on Figure 3 one of the analyzed formulas is perfect and the degree of under and overestimation is different for each formula (all plots are scattered similarly, however the maximal and minimal limits of agreement are different).

Our research provides relevant insight into CDH diagnosis and monitoring. Firstly, we demonstrated that the Hadlock formula generates reliable measurements, rendering alterations to the formula for fetal weight estimation impractical. Time of the measurements, especially proximity to the delivery, is a relevant aspect as well. Weekly fetal biometry helps minimize the discrepancy between EFW and birthweight. However, the rate of fetal growth does not justify measurements at intervals of <7 days, not to mention that more frequent assessments may broaden the error margin [22].

Our findings are clinically relevant and important as birthweight estimation may be important factor in management of neonates with the most severe forms of congenital diaphragmatic hernia. With no doubt the use of traditional and well accepted Hadlock formula is more convenient compared to tailored, complicated formulas designed only for CDH fetuses, especially if there is no obvious benefit. Fetal therapy is widely available for CDH babies nowadays [23]. The most severe cases might require the use of ECMO. Most specialists consider ECMO use for infants with weight above 2 kg and above 34 weeks gestation. Considering higher then in general population risk of lower birthweight and premature delivery due to congenital diaphragmatic hernia, it is important to calculate EFW accurately. This allows to plan further management and prepare the ECMO and surgical teams for interventions well in advance.

## 5. Conclusions

As shown above, adjusting the Hadlock formula by scan-to-delivery time remains the most precise method of fetal weight estimation among cases with CDH. Therefore, it seems that Hadlock formula is a reliable tool for EFW estimation also in fetuses with CDH prior to delivery. Secondly, we believe that establishing anatomical landmarks for fetuses with stomach and liver herniation could be useful for medical practitioners, as results across the literature are not uniform and show varying degree of discrepancy between different centers [13,20].

## Figures and Tables

**Figure 1 jcm-13-03392-f001:**
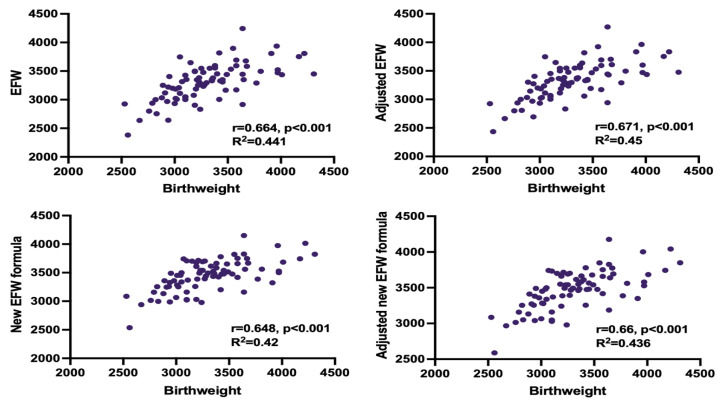
Correlation between estimated fetal weight (EFW) and birthweight for all measurement methods in healthy infants.

**Figure 2 jcm-13-03392-f002:**
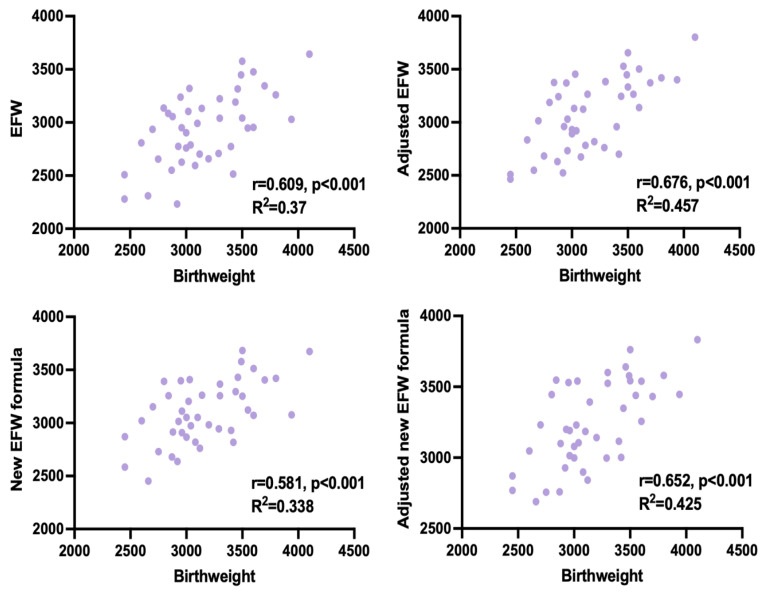
Correlation between estimated fetal weight (EFW) and birthweight for all measurement methods in CDH infants.

**Figure 3 jcm-13-03392-f003:**
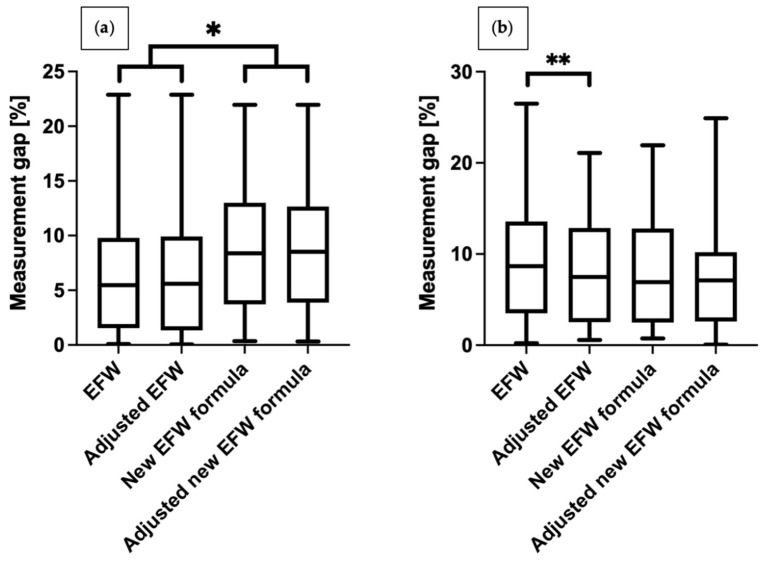
Percentage error margin between fetal weight estimated with different methods and neonatal birthweight: (**a**) healthy newborns, (**b**) newborns with CDH. * *p* < 0.0083; ** *p* = 0.007.

**Table 1 jcm-13-03392-t001:** Maternal characteristics at baseline, presented as mean ± SD.

Variable	Control Group (n = 80)	Study Group (n = 42)	*p*-Value
Age	31.3 ± 3.9	29.9 ± 4.7	NS
BMI	26.7 ± 3.9	28 ± 3.9	NS
Gravidity	1.8 ± 0.8	2.1 ± 1.1	NS
Parity	1.8 ± 0.8	1.9 ± 1.0	NS
Gestational age at delivery [weeks + days]	38 + 3	38 + 4	NS
Interval between US scan and delivery [days, median, range]	0 [0–16]	5 [0–14]	<0.001

NS—not significant. BMI—body mass index [kg/m^2^]. US—ultrasound.

**Table 2 jcm-13-03392-t002:** Descriptive characteristics of ultrasound measurements in fetuses, presented as mean ± SD or %.

Variable	Control Group (n = 80)	Study Group (n = 42)	*p*-Value
Birthweight [g]	3306.5 ± 375.8	3162.9 ± 386.3	0.049
AC [mm]	341.1 ± 15.9	320.3 ± 17.8	<0.001
AC percentile [%]	42 ± 23.7	36.6 ± 60.1	0.005
BPD [mm]	92 ± 4.1	92.5 ± 3.4	0.49
BPD percentile [%]	41.9 ± 27.6	56.1 ± 28.8	0.006
HC [mm]	328.2 ± 11	329 ± 12.5	NS
HC percentile [%]	34.2 ± 24.6	45.6 ± 30.4	NS
FL [mm]	73.4 ± 3.8	70.9 ± 2.95	<0.001
FL percentile [%]	58.8 ± 28.6	47 ± 25.5	NS
Liver up		19 (45.2%)	-
Alive at 28 days		31 (73.8%)	-
Left-side herniation		39 (92.9%)	-
FETO procedure		9 (21.4%)	-

NS—not significant. AC—abdominal circumference. BPD—biparietal diameter. HC—head circumference. FL—femur length. FETO—fetoscopic endoluminal tracheal occlusion.

**Table 3 jcm-13-03392-t003:** Descriptive characteristics of ultrasound measurements in fetuses with left-sided vs. right-sided CDH, presented as means ± SD.

Variable	Left-Sided CDH (n = 39) [Mean ± SD]	Right-Sided CDH (n = 3) [Mean ± SD]	*p*-Value
Birthweight [g]	3131.1 ± 401.4	3096.7 ± 167.4	NS
AC [mm]	320.9 ± 18.3	315.3 ± 6.9	NS
AC percentile [%]	38.3 ± 59.9	18.5 ± 20.2	NS
BPD [mm]	92.5 ± 3.3	91.7 ± 4.6	NS
BPD percentile [%]	56.6 ± 27.4	54.6 ± 44.2	NS
HC [mm]	329.1 ± 12.4	321.9 ± 2.8	NS
HC percentile [%]	46.9 ± 30.2	24.4 ± 12.7	NS
FL [mm]	70.8 ± 3.0	70.1 ± 0.6	NS
FL percentile [%]	46.4 ± 25.8	34.5 ± 11.7	NS
Liver up	17 (43.6%)	2 (66.6%)	NS
Alive at 28 days	28 (71.8%)	3 (100%)	NS

NS—not significant. AC—abdominal circumference. BPD—biparietal diameter. HC—head circumference. FL—femur length.

**Table 4 jcm-13-03392-t004:** Descriptive characteristics of ultrasound measurements in fetuses after FETO vs. no-FETO procedure, presented as mean ± SD.

Variable	FETO (n = 9)	No-FETO (n = 33)	*p*-Value
Birthweight [g]	2836.7 ± 423.9	3215.8 ± 346.2	0.008
Gestational age at delivery [weeks + days]	37 + 1	38 + 5	<0.0001
AC [mm]	312.6 ± 18.6	322.5 ± 17.3	NS
AC percentile [%]	31.8 ± 30.0	28.6 ± 21.8	NS
BPD [mm]	91.6 ± 3.9	92.6 ± 3.2	NS
BPD percentile [%]	61.9 ± 28.7	55.1 ± 28.2	NS
HC [mm]	327.2 ± 15.2	328.9 ± 11.6	NS
HC percentile [%]	57.4 ± 31.9	42.5 ± 29.0	NS
FL [mm]	69.5 ± 2.7	71.1 ± 3.0	NS
FL percentile [%]	45.7 ± 23.6	45.6 ± 25.9	NS
Liver up	5 (55.5%)	14 (42.4%)	NS

NS—not significant. FETO—fetoscopic endoluminal tracheal occlusion. AC—abdominal circumference. BPD—biparietal diameter. HC—head circumference. FL—femur length.

**Table 5 jcm-13-03392-t005:** Numerical differences between estimated fetal and actual infant weight with Pearson’s correlation coefficient for each method.

Group	Variable	Mean ± SD [g]	Mean Difference ± SD [g]	95% Confidence Interval	*p*-Value	r *
Control group	Weight	3306.5 ± 375.8				
	EFW	3304.7 ± 323.8	217.1 ± 191.9	176.7–259.2	NS	0.664
	Adjusted EFW ^1^	3326.8 ± 320.4	213.3 ± 191.5	171.3–255.3	NS	0.671
	Faschingbauer’s formula ^2^	3448.2 ± 279.7	271.2 ± 170.2	233.9–308.5	<0.001	0.648
	Adjusted Faschingbauer’s formula	3470.3 ± 274.7	279.9 ± 168.2	243.0–316.8	<0.001	0.660
Study group	Weight	3162.9 ± 386.3				
	EFW	2942.6 ± 344.5	315.3 ± 231.4	245.3–385.1	<0.001	0.609
	Adjusted EFW ^1^	3080.4 ± 345.5	248.9 ± 177.8	194.8–303.2	0.079	0.676
	Faschingbauer’s formula ^2^	3103.2 ± 297.6	260.9 ± 195.2	201.9–320.4	NS	0.581
	Adjusted Faschingbauer’s formula	3241 ± 301.6	244.2 ± 182.9	189.4–300.5	NS	0.652

^1^ Adjusted EFW = (EFW + 185*time gap[days]/7). ^2^ Faschingbauer’s formula = EFW = 10^(4.6729 107 371 + 0.2365 011 768 * HC + 0.2228 897 682 * FL^2 − 0.0129 895 773 * FL^3 − 1.0470 039 072 * (FL * HC)^0.5 + 0.0004 314 661 * (AC * HC) − [in case of liver herniation] 0.0062 112 122)(12). * All calculated coefficients of determination had statistical significance of *p* < 0.001. NS—not significant.

**Table 6 jcm-13-03392-t006:** Average of individual differences between estimated fetal weight and actual birthweight.

	Group	Mean ± SD	Minimum–Maximum	*p*-Value
EFW % gap	Control group	6.5 ± 5.4%	0.09–22.9%	0.005
	Study group	9.7 ± 6.6%	0.2–26.5%	
Adjusted EFW % gap	Control group	6.4 ± 5.5%	0.06–22.9%	NS
	Study group	7.8 ± 5.5%	0.6–21.1%	
Faschingbauer’s formula % gap	Control group	8.4 ± 5.4%	0.4–21.9%	NS
	Study group	8.2 ± 5.9%	0.7–21.9%	
Adjusted Faschingbauer’s formula % gap	Control group	8.7 ± 5.4%	0.3–21.9	NS
	Study group	7.9 ± 6.5%	0.1–24.9%	

NS—not significant. EFW—estimated fetal weight.

**Table 7 jcm-13-03392-t007:** Percentage differences between estimated fetal weight and actual birthweight of study group.

Group	Mean ± SD	Minimum–Maximum	*p*-Value
EFW % gap	9.7 ± 6.6%	0.2–26.5%	0.0068
Adjusted EFW % gap	7.8 ± 5.5%	0.6–21.1%	NS
Faschingbauer’s formula % gap	8.2 ± 5.9%	0.7–21.9%	0.69
Adjusted Faschingbauer’s formula % gap	7.8 ± 5.5%	0.6–21.1%	NS
EFW % gap	9.7 ± 6.6%	0.2–26.5%	0.094
Faschingbauer’s formula% gap	7.9 ± 6.5%	0.1–24.9%	NS
Adjusted EFW % gap	7.8 ± 5.5%	0.6–21.1%	0.88
Adjusted Faschingbauer’s formula % gap	7.9 ± 6.5%	0.1–24.9%	NS

NS—not significant, EFW—estimated fetal weight.

**Table 8 jcm-13-03392-t008:** Percentage differences between estimated fetal weight and actual birthweight of control group.

Group	Mean ± SD	Minimum–Maximum	*p*-Value
EFW % gap	6.5 ± 5.4%	0.09–22.9%	0.45
Adjusted EFW % gap	6.4 ± 5.5%	0.06–22.9%	NS
Faschingbauer’s formula % gap	8.4 ± 5.4%	0.4–21.9%	0.07
Adjusted Faschingbauer’s formula % gap	8.7 ± 5.4%	0.3–21.9	NS
EFW % gap	6.5 ± 5.4%	0.09–22.9%	0.0016
Faschingbauer’s formula% gap	8.4 ± 5.4%	0.4–21.9%	NS
Adjusted EFW % gap	6.4 ± 5.5%	0.06–22.9%	<0.001
Adjusted Faschingbauer’s formula % gap	8.7 ± 5.4%	0.3–21.9%	NS

NS—not significant, EFW—estimated fetal weight.

## Data Availability

All data generated or analyzed during this study are included in this article. Further enquiries can be directed to the corresponding author.

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
