# Peer review of "Accuracy of Estimated Fetal Weight Assessment in Fetuses with Congenital Diaphragmatic Hernia—Is the Hadlock Formula a Reliable Tool?"

_jcm, 2024, doi:10.3390/jcm13123392_

Round 1

Reviewer 1 Report

Comments and Suggestions for Authors

Article` ``Accuracy of estimated fetal weight assessment in fetuses with congenital diaphragmatic hernia – is the Hadlock formula a re-

liable tool?``

The research is done to compare the accuracy of Hadlock's formula and Faschingbauer's formula for fetal weight estimation in the CDH fetus population. In this study, we demonstrated that the Hadlock formula is a reliable method for calculating birth weight in infants with CDH, if adjusted for the time interval between the ultrasound measurements and delivery. The use of a novel, tailored approach proposed by Faschingbauer et al., showed no definite advantage over Hadlock within our study sample.

dealing with the subject of CDH and body weight assessment, the authors deal more with statistics than with the applicability of the method for the outcome and help in the diagnosis, as well as the treatment of CDH

In the introduction, it is better to describe the causes of CDH, and not in the discussion, to present the results more adequately, here it is as if I am reading poorly processed statistics, a tabular or only fractional presentation is better than a dispersion

Reformulate the CEO paper

Author Response

Dear Reviewer, 

We would like to thank you for evaluating our manuscript. Your suggestions raise important issues and their inputs are very helpful in improving the manuscript. We agree with the comments and we have revised our manuscript accordingly. Please see a detailed list of answers below. The new corrected version of the manuscript is attached. All changes are marked in colour in the text.  We also provide a clean version of the manuscript.

To make this manuscript more clinically relevant and to simplify the statistics we have applied amendments according to your suggestions:

  1. More clinical information on CDH was added to introduction including anatomical description and prognosis.
  2. There is no information on CDH causes in the discussion section.
  3. A section on clinical application of this research was added at the end of the discussion
  4. To simplify the statistics we have deleted Bland-Altman plots, which were distracting and were not the major findings of this research.   
  5. We agree statistics are easier to read and understand if results are presented in tables or described in fractions. Therefore now in our manuscript there are 8 tables and only 3 pictures. 

 Hope you will find our corrections satisfying and the manuscript suitable for publication. 

Reviewer 2 Report

Comments and Suggestions for Authors

The Authors tackle a problem of underestimating fetal birth weight, as measured by ultrasound, in cases of congenital diaphragmatic hernia. In this study they compared the accuracy of Hadlock’s formula and Faschingbauer’s formula for fetal weight estimation in a large group of 42 fetuses with congenital diaphragmatic hernia, and in 80 healthy matched controls.

The estimated fetal weight adjusted for the interval between the ultrasound and delivery had a good correlation in both a study group and in controls. Moreover, the Hadlock formula predicted birth weight in congenital diaphragmetic hernia casas with a 7.8 ±5.5% error and the Faschingbauer’s formula with a 7.9 ± 6.5% error, respectively. This study underlines a good predictive value of the scan-to-delivery time adjusted Hadlock formula as ultrasound tool estimating fetal weight in cases of congenital diaphragmatic hernia.

I find this article intersting, well concepted and miticulously described. It may be important for every-day prenatal diagnosing.

Author Response

Dear Reviewer, 

We would like to thank you for your time and for evaluating our manuscript. We appreciate your expertise and experience. We do hope the new version of the manuscript is even more clinically relevant and interesting. 

Round 2

Reviewer 1 Report

Comments and Suggestions for Authors

accetable in its current form